# Incorporating Ceragenins into Coatings Protects Peripherally Inserted Central Catheter Lines against Pathogen Colonization for Multiple Weeks

**DOI:** 10.3390/ijms241914923

**Published:** 2023-10-05

**Authors:** Aaron Zaugg, Elliot Sherren, Rebekah Yi, Tessa Larsen, Brayden Dyck, Sierra Stump, Fetutasi Pauga, Anna Linder, Meg Takara, Emily Gardner, Spencer Shin, Jace Pulsipher, Paul B. Savage

**Affiliations:** Department of Chemistry and Biochemistry, Brigham Young University, Provo, UT 84602, USAbekahyi24@gmail.com (R.Y.); megtakara@gmail.com (M.T.);

**Keywords:** PICC lines, device colonization, healthcare-acquired infection, biofilm, ceragenins, bacteria, fungi, multi-drug resistance, polyurethane

## Abstract

Healthcare-acquired infections and multi-drug resistance in pathogens pose a major crisis for the healthcare industry. Novel antibiotics which are effective against resistant strains and unlikely to elicit strong resistance are sought after in these settings. We have previously developed synthetic mimics of ubiquitous antimicrobial peptides and have worked to apply a lead compound, CSA-131, to the crisis. We aimed to generate a system of CSA-131-containing coatings for medical devices that can be adjusted to match elution and compound load for various environments and establish their efficacy in preventing the growth of common pathogens in and around these devices. Peripherally inserted central catheter (PICC) lines were selected for our substrate in this work, and a polyurethane-based system was used to establish coatings for evaluation. Microbial challenges by methicillin-resistant *Staphylococcus aureus*, *Pseudomonas aeruginosa*, *Klebsiella pneumoniae*, and *Candida albicans* were performed and SEM was used to evaluate coating structure and colonization. The results indicate that selected coatings show activity against selected planktonic pathogens that extend between 16 and 33 days, with similar periods of biofilm prevention.

## 1. Introduction

An ongoing problem for hospitals and other healthcare facilities is the threat of nosocomial or healthcare-acquired infections (HAIs). According to the World Health Organization’s (WHO) 2022 report on infection prevention and control, 7–15 hospitalized patients in 100 will acquire at least one HAI, with 1 in every 10 infected patients dying from the infection [1]. Notably, during and since the COVID-19 pandemic, HAIs have been on the rise, and due to increasing antibiotic resistance, infections are becoming more dangerous [2]. Antibiotic resistance is a worsening global crisis arising from the use and misuse of antibiotics and lack of newer drugs, and as many as 46% of bacteria from healthcare or other hygienic facilities are multi-drug resistant (MDR) [3,4]. Compounding this problem is the prevalence of medical-device-related infections. For example, despite the use of antibiotics, as many as 2.8% of patients using a peripherally inserted central catheter (PICC) line suffer bloodstream infections [5,6].

Given the severity of MDR infections in healthcare outcomes, efforts have been made to obtain novel antibiotics to combat current resistance patterns, and an attractive starting point is the consideration of endogenous mechanisms for controlling microbial growth. Cationic antimicrobial peptides (AMPs) are one such natural source of pathogen control, utilizing electrostatic and hydrophobic interactions to permeabilize membranes and allow the leakage of intracellular components of bacteria and fungi [7]. These AMPs are ubiquitous in multicellular eukaryotes, and yet they remain effective at controlling invasive pathogens, indicating that their mechanisms are unlikely to induce strong mechanisms of resistance [7,8]. The investigation of these peptides as therapeutics has shown promise, but there are substantial obstacles to their application, such as difficulty of large-scale synthesis, hydrolytic degradation, rapid system clearance, and thermal instability [9].

Our group has designed and synthesized ceragenins as small-molecule, non-peptide mimics of the amphipathic nature of AMPs and their membrane-disrupting mechanisms while avoiding many of the obstacles of utilizing AMPs. The second generation of ceragenins have been tested recently on MDR strains, including *Klebsiella pneumonia* and *Pseudomonas aeruginosa*, with ceragenin CSA-131 standing out as a lead compound [10,11]. As a mimic of AMPs, ceragenin CSA-131 associates with bacterial membranes, and due to its amphipathic morphology, this association results in membrane depolarization. With Gram-positive bacteria, the ceragenin gains access to the cytoplasmic membrane directly; with Gram-negative bacteria, CSA-131 traverses the outer membrane to access the cytoplasmic membrane. Further work has shown that CSA-131 does not suffer from cross resistance with other membrane-active antimicrobials such as chlorhexidine and colistin [12]. Due to the efficacy of CSA-131 against MDR strains of high concern, an attractive application of the technology may be found in preventing the colonization of medical devices. The relatively low cost of the production of CSA-131 and its stability offer the possibility of incorporating it into processes which require high temperatures and long storage times, which leverage its advantages over AMPs and other antimicrobials. We thus aimed to explore its compatibility with polymer coatings on medical devices, specifically PICC lines. Our objective was to reduce the percentage of patients that suffer with bloodstream infections associated with PICC lines by applying a coating that inhibits microbial colonization for an extended period.

Multiple approaches have been explored to inhibit the microbial colonization of catheters, and these, in general, employ multiple antimicrobials/antibiotics to provide a spectrum of activity that encompasses Gram-positive and -negative organisms [13,14,15]. These combinations include antibiotics to which resistant bacteria have been identified and meta-analysis has identified problematic bacterial or fungal strains for all tested combinations [15]. Due to the breadth of the spectrum of ceragenins, CSA-131 may be used as a single antimicrobial to prevent colonization by Gram-positive and -negative bacteria and fungi. Furthermore, as a mimic of AMPs, ceragenins are unlikely to engender resistance. Consequently, CSA-131-containing coatings are well suited as “stand-alone”, long-term solutions to the microbial colonization of catheters.

The process of effectively incorporating an antimicrobial active into a coating can be complicated by several factors, including chemical reactivity of the antimicrobial with the coating material, poor adherence of the coating to the substrate, and rapid loss of the antimicrobial from the coating [16]. CSA-131 contains multiple amine groups, which can be reactive with monomers and prepolymers from which polymeric coatings are generated. Furthermore, these amine groups make CSA-131 highly water-soluble, allowing for rapid elution from a coating. To avoid these issues, we developed a naphthalene disulfonate salt form of CSA-131. This salt form, CSA-131NDSA, is sparingly soluble in water; ion exchange is required to allow CSA-131 to become highly water-soluble. This salt exchange process can control the release of CSA-131 from a coating [17]. In addition, in the CSA-131NDSA salt form, the amine groups are not reactive, thus preventing them from inhibiting polymerization reactions. Thus, CSA-131NDSA presents favorable attributes for incorporation into a polymeric coating for medical devices.

## 2. Results

### 2.1. CSA-131NDSA Can Be Stably Integrated into Polyurethane Coatings

Our goal was to develop a simple dip-coat system to generate coatings to prevent colonization by common pathogens, and we selected PICC lines as an initial substrate due to the prevalence of infections associated with this medical device. PICC lines are inserted an estimated 2,500,000 times a year worldwide with a rate of infection as high as 2.92% [18]. We set out to identify a suspension of monomers that would result in a polymerized coating on the surface of a PICC line. We chose to use a commercially available system that generates polyurethane hydrogels on medical devices. By forming the polymer on the medical device surface after dip-coating, the resulting polymer adheres well to the surface. Furthermore, polyurethanes have excellent biocompatibility, mechanical stability, and once inserted into a patient, they soften, resulting in increased patient comfort [19]. By differing the percentages of solids in the coating solution (methylethylketone solvent), coating thicknesses were controlled, giving coatings which would adhere to the surface in both dry and wet conditions. We also determined that the inclusion of an initial silicone primer on the PICC line segments reduced variability in mass changes associated with the dip-coating process, and all samples were subsequently primed. By suspending CSA-131NDSA salt at 20% (*w*/*w*) of total solids in the urethane solution, we arrived at our first coating system (Coating A). The initial results were promising, with the coatings providing a 3-log reduction in colony forming units (CFUs) for three days with daily challenges due to methicillin-resistant *Staphylococcus aureus* (MRSA) (Figure 1A).

### 2.2. Multi-Layer Coatings Can Control Total CSA-131 and Elution

We next aimed to increase the CSA-131 reservoir to expand potential applications to more rigorous conditions which may exist in vivo. By assaying concentrations of both urethane prepolymers and CSA-131NDSA, we arrived at a second system that contained 5% (*w*/*v*) urethane prepolymers with CSA-131NDSA at 50% (*w*/*w*) of total solids (Coating B). The increased concentrations and viscosity resulted in a nearly 12-fold increase in CSA-131 in the coating compared to our earlier system (Figure 1B). The elution profile for this coating, however, indicated that we were losing over half of the load of CSA-131 within 24 h of exposure to an aqueous environment, and visible swelling of the coating was observed, indicating that the ratio of solids in the coating did not allow for sufficient crosslinking between urethanes for stability (Figure 1C).

Aiming to conserve the higher reservoir of CSA-131NDSA, reduce initial elution, and increase stability, we investigated the use of a top coat. After curing the initial coating, segments were immersed in the same 5% urethane prepolymer solution we had previously made without CSA-131NDSA. After curing, the layered coatings led to a system (Coating C) that had no visible swelling in water, had a lower initial release of CSA-131, and a more sustained release on subsequent days (Figure 1C).

### 2.3. Polyurethane Coating Has Consistent, Distinct Layers Fully Covering the PICC Line Surface

While the early development of our coating system had largely been guided by changes in mass and elution, we wished to visualize coated segments to gauge coating uniformity and thickness. We utilized scanning electron microscopy (SEM) to visualize Coating C, and initial observations of the surface of coated and uncoated PICC line segments verified that the coating left no exposed tubing (Figure 1D,E). Questions remained on the consistency and nature of the internal structure of the coating, so a freeze–crack method was used to expose the layers of the combined coating.

Analysis of the cross-section of the coating revealed a structure consisting of several distinct layers (Figure 1F). The first layer, approximately 4 µm thick, directly adjacent to the substrate, is the silicone primer layer. The next layer, which is approximately 18 µm thick, is the base polyurethane layer containing CSA-131NDSA. Note that there is a clear boundary between the primer layer and the base coating layer, due not only to the use of differing polymers during the formation of these layers, but also to the inclusion of CSA-131NDSA in the base coating. The top polyurethane layer of the coating, which is approximately 12 µm thick, is also clearly visible on the SEM image. Comparison of this layer and the underlying, CSA-131NDSA-containing layer shows a greater amount of heterogeneity in the lower layer, which likely allows for a greater amount of solvent infiltration. The homogeneity of the top layer was expected to restrict solvent penetration and delay CSA-131 elution.

### 2.4. Coated PICC Lines Reduce Planktonic Growth of Bacterial and Fungal Pathogens

To test the antimicrobial efficacy of Coating C, we selected representative Gram-positive and Gram-negative bacterial and fungal strains, specifically MRSA, *P. aeruginosa* (PA), *K. pneumoniae* (KP), and *C. albicans* (CA). These were selected as they are responsible for a majority of central-line-associated bloodstream infections (CLABSIs) identified in clinical studies [20,21]. Further, all have been designated as high- or critical-priority pathogens by the WHO [22,23].

In this study, PICC line segments were exposed daily to the inocula of the indicated pathogen in the appropriate fresh growth medium. We found that PICC line segments with Coating C gave multiple weeks of at least a 3-log reduction in pathogenic growth. Against MRSA, the first growth in the medium appeared at day 26, and a total of 33 days of statistically significant reduction in bacterial growth was observed (Figure 2A). With PA, growth was first detected at day 13, and 16 days of statistically significant reduction were observed (Figure 2B). Coating C performed similarly with the other Gram-negative strain, KP, with first growth at 14 days, and statistical significance reduction throughout day 16 (Figure 2C). For CA, growth was observed at day 17, and significant reduction occurred throughout day 21 (Figure 2D).

At later stages in the studies shown in Figure 2, substantial variation was measured, as indicated by larger error bars. Small variations in coating thicknesses among samples may lead to small differences in CSA-131 elution. When concentrations of CSA-131 generated are near the MICs of the targeted organisms, substantial variations in the total growth are more likely. While significant reductions in growth from pathogens of concern for multiple weeks remain a promising benchmark, these assays have described coating activity against planktonic forms of the microorganisms; an even greater indicator of potential success is the prevention of biofilm formation.

### 2.5. Coated PICC Lines Prevent Biofilm Formation

Biofilms form when pathogens secrete matrix materials, which encase a community of microbes that adhere to tissue or surfaces [24]. These biofilms can release planktonic cells and/or additional biofilm that can infect other tissues. Biofilm is notorious in its ability to continue to grow in stressful environments. One study found that microorganisms in biofilm form to grow in the presence of antibiotics 1000 times more concentrated than the concentration needed to kill off planktonic cells [25]. This inherent resistance can lead to repeated infections as biofilm persists even after administered antibiotics kill planktonic cells. Due to the significance of biofilm in infection, we measured the ability of Coating C on PICC line segments to prevent biofilm colonization.

Coated (Coating C) and control PICC line segments were prepared and challenged daily with fresh inocula in the appropriate medium. To determine the number of microbial cells adhered to the segments, at designated time points, segments were removed from the medium, gently washed with phosphate-buffered saline to remove non-adhered cells, and sonicated in a neutralizing medium to release cells from the biofilm. With bacteria, counts within established biofilms on uncoated PICC line segments were 6–8 logs after one day of incubation (Figure 3A–C), and with CA, biofilm counts were 4 to just over 6 logs (Figure 3D). Counts remained in these ranges even after extensive reinoculation and incubation. Segments with Coating C were protected for extended periods from colonization. For example, with MRSA, after a single day of incubation, the number of bacterial counts on coated segments was below the detection limit (1 log), and counts remained at this level for 29 days of repeated inoculations and growth medium exchanges (Figure 3A). By day 34, bacteria were isolated from the coated segments; however, as compared to the control, the coating reduced the biofilm by more than 3 logs. With the other organisms, Coating C protected the segments for at least 14 days (Figure 3B–D).

We additionally sought to explore if biofilm formation was completely prevented or if some biofilm formation occurred and was eliminated within our sampling window. In order to visualize biofilm on coated segments, we used SEM and samples challenged with PA. Comparing the unchallenged PICC line surface (Figure 1D) and the segment challenged by PA for 7 days (Figure 3E) demonstrates the ability of PA to establish a mature biofilm on the uncoated surface. A sheet of matrix proteins has been established, and individual rod-shaped cells can be seen both under the surface and partially exposed, potentially representing cells about to be shed from the biofilm. Conversely, comparing the unchallenged coating (Figure 1E) and the challenged coating (Figure 3F) shows the surface free of the established protein matrix and rod-shaped cells. Small aggregates can be observed on the surface, which may represent the cellular remnants of bacteria, but no evidence of biofilm formation was observed.

### 2.6. Coated PICC Lines Retain Partial Efficacy against Pathogens in High-Protein Environments

Bacterial and fungal growth media typically do not contain high concentrations of protein. However, implanted medical devices may be exposed to high protein concentrations, and protein deposition on surfaces can influence microbial adhesion. In addition, the antimicrobial efficacy of antibiotics can be influenced by high protein concentrations, and protein aggregation has been observed as a mechanism of antibiotic resistance [26]. Therefore, we repeated the planktonic efficacy assays in a high-protein environment. We prepared media containing 70 mg/mL serum protein supplemented with standard growth media to challenge PICC line segments. In the presence of coated (Coating C) segments, MRSA growth was significantly reduced for 10 days (Figure 4A), while with the Gram-negative pathogens, microbial growth was observed on day 4 (Figure 4B,C). The coated segments significantly inhibited the growth of CA for five days (Figure 4D).

## 3. Discussion

CLABSI and HAI in general have been growing concerns for decades, complicated by drug resistance that has accompanied the widespread use and misuse of antibiotics. CSA-131 has previously been shown to not be impacted by cross-resistance to other antibiotics and is effective against drug-resistant isolates [10,11,12], making it a strong potential candidate for alleviating the current MDR crisis. We have demonstrated the ability to incorporate CSA-131 into a simple urethane-based dip-coating system at varying concentrations and thicknesses.

By varying the number of layers and the concentrations of urethane prepolymer and CSA-131NDSA in the dipping suspension, we optimized the coating thicknesses, the CSA-131NDSA load, and CSA-131 elution. The ease of modulating the characteristics of the final coating by simple concentration adjustments allows a broad range of coating types to be derived from this work to match desired elution amounts and duration.

The results from standard microbiological assays demonstrate that Coating C can prevent local planktonic growth for at least two weeks for representative strains of Gram-positive, Gram-negative, and fungal pathogens of critical or high concern [22,23]. More importantly, a 3-log reduction in biofilm colonization of the device segments was achieved for similar periods. The prevalence of biofilm in HAI is around 65%, and biofilms have been shown to facilitate genetic distribution, facilitating the spread of MDR genes [27]. As such, devices which protect from biofilm formation even after insertion could help reduce not only infections but the spread of MDR patterns between different strains. These concerns are not limited to just PICC line infections, and the coating system we have developed could be utilized in a much broader range of medical device applications.

Of note, the Gram-negative bacteria, KP and PA, were the first to overcome our devices in all assays, but this is expected because Gram-negative bacteria have two membrane layers encapsulating a periplasm, which allows them to better regulate material uptake from the surrounding environment [28]. They also have efflux pumps to identify and extrude noxious elements from the periplasm, and they have been demonstrated under stress to downregulate the number of porin channels in their outer membrane, all of which reduces the ability of antibiotics to enter the outer membrane and access the inner membrane [29,30,31]. Such characteristics of Gram-negative bacteria differentiate them from Gram-positive bacteria and contribute to Gram-negative bacteria having a higher resilience, especially for membrane-active compounds, which is consistent with our results. To improve activity against Gram-negative targets, the synergistic activity of ceragenins with other antibiotics is already being explored [32,33].

The in vitro work described above demonstrates the ability of the optimized coating (Coating C) to control the release of CSA-131 for an extended duration, as evidenced by the antimicrobial properties of the coating. These studies include repeated inoculations with rapidly growing organisms, and because the growth media were exchanged on a daily basis, CSA-131 could not accumulate; the only antimicrobial activity came from CSA-131 eluting within a 24 h time frame. Similarly, in vivo biological fluid exchange in most environments will preclude the accumulation of an eluting active antimicrobial. However, the magnitude of microbial inoculation will likely be lower in vivo than in these studies. While large populations of microorganisms may be introduced upon device implantation, further exposure to large numbers of microorganisms is less likely.

The presence of high protein loads decreased the duration of antimicrobial activity of Coating C; nevertheless, MRSA growth was significantly inhibited for nine days, which is particularly important because bacteremia and endocarditis due to *S. aureus* is a growing concern [34]. Additionally, another recent study of coatings on polyurethane catheters used blood conditioning and found that MRSA reduction even after 3 days was enough to yield beneficial in vivo results. In that study, on a benzophenone-based amide coating, though the in vitro reduction in MRSA was only observed for 3 days, in a mouse model, the device achieved a 99.9% reduction in MRSA growth [35]. Further, the conditions of our assay may oversimplify antimicrobial activity in blood. Circulating blood contains immune factors which can synergize with or be recruited by antimicrobial substances, including LL-37, which could lead to improved performance in vivo relative to our assays [36,37].

While additional research needs to be conducted to confirm that the combination of therapeutic, device, and polymer will not present novel risks, the individual components are all being used within medical applications already. Polyurethanes have been approved by the United States Food and Drug Administration for use in multiple devices, including PICC lines, and have well-documented safety data [38]. CSA-131 has also been previously integrated into nebulizers and devices for human use in products at various stages of human clinical trials [39]. The emphasis of this work has been on combining these components to craft adjustable systems, and while safety testing in both in vitro and animal models will need to be performed in the future, we are optimistic that our coatings can be formulated for safe applications.

## 4. Materials and Methods

### 4.1. Preparation of PICC Line Segments

Single lumen BARD PowerPICC™ 4F PICC lines (Lucent Medical Systems, Kirkland, Brentwood, TN, USA) were used in this study. Lines were cut into 15 mm sections and cauterized to isolate a single accessible side for preparation and testing. Segments were mounted on 22Ga needles to facilitate handling before being primed as advised by the manufacturer (Hydromer 2314-172, Somerville, NJ, USA) and cured for 8 h at 70 °C. Urethane prepolymer solution (Hydromer 2018-20M) was acquired at 3% *w*/*v* and was adjusted to target concentrations through solvent removal using a Buchi R-100 Rotavapor System. For coating layers with CSA-131NDSA, powder was weighed, and polyurethane solution was added to acquire the target CSA-131NDSA percent of total solids; the final suspension was sonicated at 0 °C for 4–8 h to achieve complete dispersal of the solids. PICC line segments were submerged in the suspension for 5 s and moved to an oven to cure overnight at 90 °C.

### 4.2. Microbial Cultures

All samples were cultured in trypticase soy broth (TSB) for bacteria and Emmons modified sabouraud dextrose broth (EMSDB) for fungi. All pathogens tested are prepared by introducing single colonies from fresh culture plates into culture media and allowed to grow overnight on an orbital shaker at 30 °C. Overnight cultures were washed and centrifuged three times in phosphate-buffered saline (PBS) and resuspended in PBS. Optical density (OD) readings were acquired at 600 nm on a Genesys 30 spectrophotometer. Aliquots of bacterial cultures were diluted to 10^6^ CFU/mL in 10% TSB or 70% fetal bovine serum (FBS) with 10% TSB for high-protein testing. Aliquots of fungal cultures were diluted to 10^3^ CFU/mL in 10% EMSDB or 70% fetal bovine serum (FBS), with 10% EMSDB for high-protein testing. The rationale for using 70% FBS is that whole-blood protein is estimated at 70 mg/mL, and serum alone contains approximately 100 mg/mL protein [40,41].

### 4.3. Total Extraction and Elution of CSA-131

Extraction and elution assays were run in triplicate. To obtain total extraction of CSA-131, PICC line segments were submerged in a solution comprised of 80% isopropanol and 20% 1 *N* HCl and heated to 70 °C for 8 h. Segments were transferred to new vessels and incubated in additional extraction solution at room temperature. Subsequent extraction steps continued at room temperature until CSA-131 peaks were indetectable using mass spectrometry. Daily elution samples were acquired from PICC line segments by incubation in PBS at 37 °C. PBS was exchanged daily at the same time, and collected samples were run on the same day. Deuterated reagents had previously been used to synthesize CSA-131D_25_, which served as an internal standard for quantification via mass spectrometry on a 6230 TOF LC/MS (Agilent Technologies, Santa Clara, CA, USA).

### 4.4. Pathogenic Challenge with Quantification of Planktonic Growth

PICC line segments were placed in culture tubes and immersed in 700 µL of inocula and incubated at 30 °C for 24 h for bacteria or 37 °C for 24 h for fungi. Microbial growth was measured by removing aliquots into Dey–Engley Neutralizing Broth (Sigma-Aldrich, St. Louis, MO, USA). The resulting suspensions were diluted and spread on nutrient agar (TSA for bacteria and EMSDA for fungi, laboratory-prepared material). Plates were incubated for 24 h before colonies were counted. Tests were run until statistical significance was lost (*p* < *0*.05).

### 4.5. Coatings Cross-Section Preparation

The cross-section of coatings was studied using the scanning electron microscope Apreo C (Thermo Fisher Scientific, Waltham, MA, USA). Standard methods for preparing cross-section (polishing, cutting) are not suitable for studying coatings (especially multilayer ones) obtained on soft polymer samples, which are associated with the transfer of particles between layers and, as a result, blurring of the boundaries between them. In connection with the foregoing, the following methodology was used in this work. At the first stage, the coated samples were frozen via immersion in liquid nitrogen (from –195 to –200 °C) and exposure for at least 15 min. After this procedure, the fragility of the sample increased significantly, and the frozen samples were cracked by bending.

### 4.6. Sample Preparation for Surface Study via Scanning Electron Microscopy

PICC line segments were prepared as described above. Segments designated for the biofilm challenge were incubated for seven days as described above. After incubation, the segments were washed with Sorensen buffer (0.1 M, pH 7.2) and then fixed in 2.5% (*w*/*v*) glutaraldehyde in Sorensen buffer at 4 °C overnight, rinsed with Sorensen buffer, immersed in an osmium tetroxide solution (0.5 mL) for 2–3 h, and washed with Sorensen buffer to remove excess osmium tetroxide. An ethanol series from 10% to 100% and hexamethyldisilazane were used to dehydrate the samples, which were then placed in an Autosamdri-931 critical point drier (Tousimis Research Corporation, Rockville, MD, USA) overnight. The samples were sputter-coated with approximately 20 nm of a gold–palladium alloy using a Quorum Q 150T ES (Electron Microscopy Sciences, Hatfield, PA, USA), and the surfaces of the PICC line segments were visualized under an Apreo C microscope (Thermo Fisher Scientific, Waltham, MA, USA) [42] (pp. 139–145).

### 4.7. Quantification of Biofilm Growth

For each pathogen, 30 coated and 30 control PICC line segments were randomly divided into 10 triplicates of each type, which were inoculated with cultures of the indicated pathogens. Every 24 h, the existing growth medium was removed, the devices were rinsed three times with 1 mL of PBS, transferred to clean culture tubes, and reinoculated in fresh growth medium. Samples were removed at predetermined intervals until significant growth was observed, defined as 15 colonies/plate, at which point samples were removed daily. Biofilm growth was quantified by removing a control and coated triplicate, which were rinsed twice with PBS. Selected segments were transferred to a culture tube and rinsed a final time to remove planktonic cells. Neutralizing broth (1 mL) was added to the tubes, which were moved to a bath sonicator (FS60, 42 kHz 100 W Thermo Fisher Scientific, Waltham, MA, USA) for 15 min, to disrupt biofilm. The segments were vortexed before samples were taken, serially diluted, and plated on agar (TSA for bacteria, EMSDA for fungi). Plates were incubated for 24 h and colonies were counted.

## 5. Conclusions, Limitations, and Future Directions

We have developed an adjustable system of components to coat soft medical devices for the prevention of microbial colonization and to reduce healthcare-acquired infections. Multiple coatings derived from this system have been evaluated on PICC line segments, and additional substrates will be explored in the future. We used a novel AMP mimic, CSA-131NDSA. This antimicrobial, in the NDSA salt form, is an ideal candidate for the prevention of the microbial colonization of medical devices: it is a broad-spectrum antimicrobial, including showing activity against fungal and drug-resistant pathogens, it is unlikely to induce microbial resistance, and its release from a hydrogel can be effectively controlled. This control is possible due to the salt exchange requirement for CSA-131 solubility and the presence of a hydrogel top coat that slows elution. This system can be applied in a few dip-coating steps and is amenable to the large-scale production of coated devices. Future work will involve the use of this system with additional devices and establishing the safety of coated devices for clinical use.

## Figures and Tables

**Figure 1 ijms-24-14923-f001:**
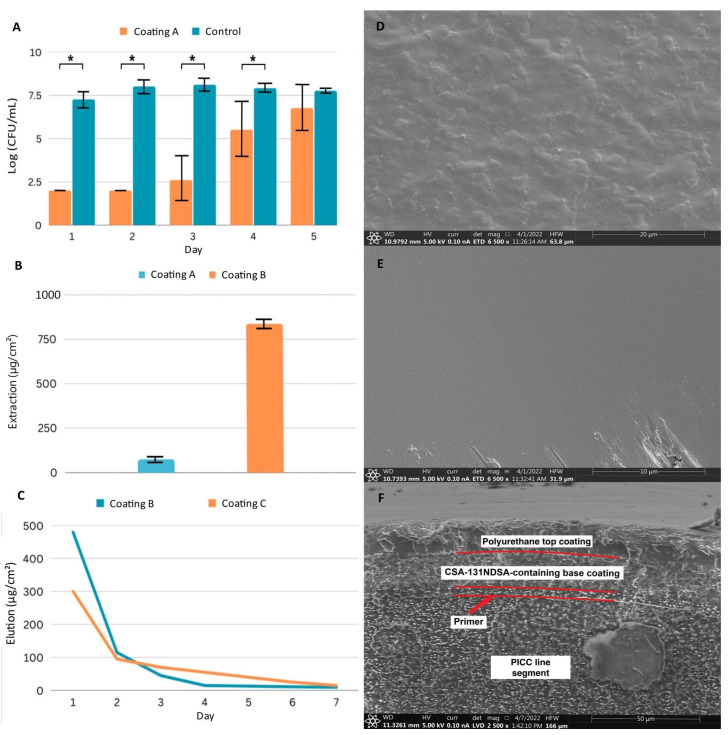
Characterization of three coating systems: Coating A samples have a 6 µm polyurethane coating containing 20% (*w*/*w*) CSA-131NDSA; Coating B samples have an 18 µm polyurethane coat containing 50% (*w*/*w*) CSA-131NDSA; Coating C samples have an 18 µm polyurethane base coat containing 50% (*w*/*w*) CSA-131NDSA with a 12 µm polyurethane top coat. (**A**) Antimicrobial efficacy assay of Coating A against repeated inoculations with MRSA. Experiments run in triplicate. * *p* < 0.05 (Student’s T test). (**B**) Total extraction of CSA-131 from triplicates of Coating A and Coating B. (**C**) Representative elution profiles of CSA-131 from Coating B and Coating C. (**D**) SEM image taken of uncoated PICC line surface. (**E**) SEM image taken of a PICC line surface coated with Coating B. Defects from handling, located along the lower edge, were included to facilitate focusing. (**F**) SEM image of PICC line cross section of Coating C. Boundaries between coating layers are visible and highlighted, while layers are labeled for clarity. Error bars represent standard deviation.

**Figure 2 ijms-24-14923-f002:**
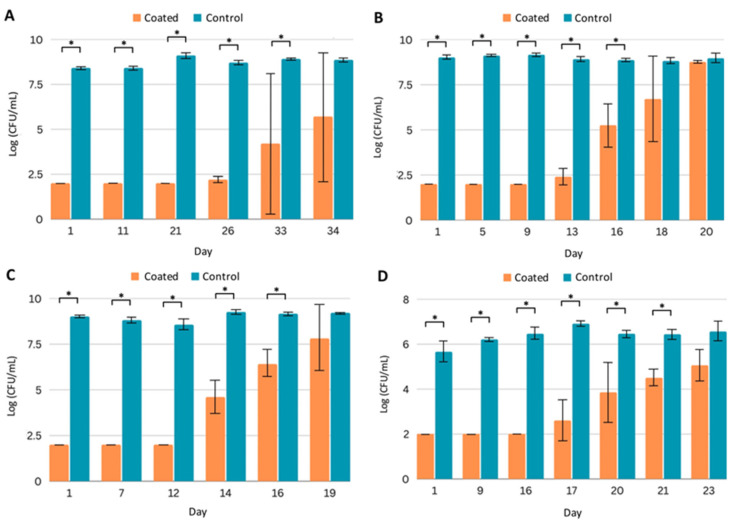
Efficacy of PICC line segments with Coating C against planktonic pathogens. Quantification of planktonic pathogen after daily challenges to uncoated (control) and coated PICC line segments by (**A**) MRSA, (**B**) *P. aeruginosa*, (**C**) *K. pneumoniae*, and (**D**) *C. albicans*. All experiments were run in triplicate. Error bars represent standard deviation. * *p* < 0.05.

**Figure 3 ijms-24-14923-f003:**
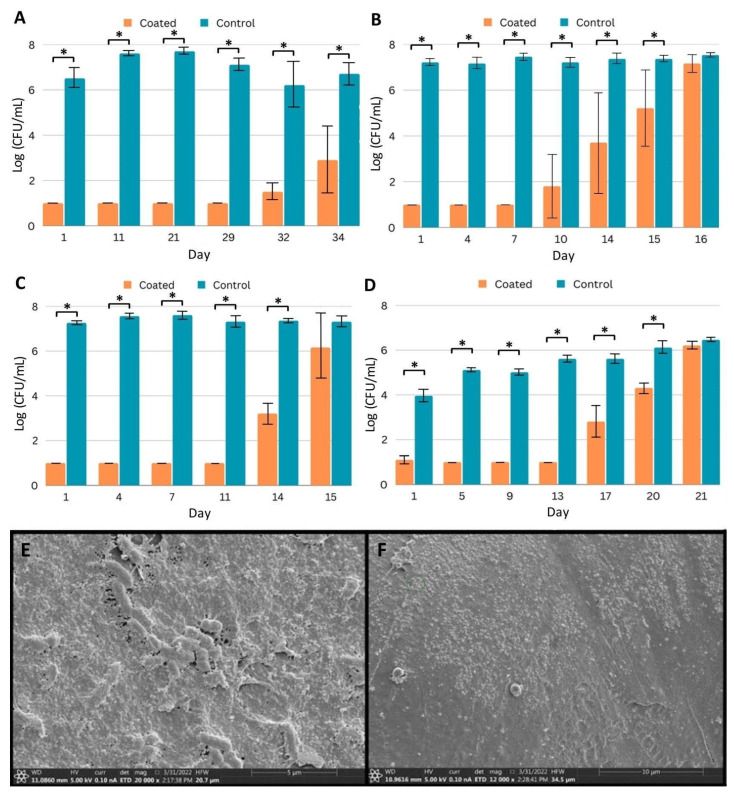
Efficacy of coated PICC line segments against biofilm formation. (**A**–**D**) Quantification of pathogen recovered from biofilm after daily challenges to control and coated (Coating C) PICC line segments by (**A**) MRSA, (**B**) *P. aeruginosa*, (**C**) *K. pneumoniae*, and (**D**) *C. albicans*. Samples were challenged with fresh inocula and media daily, and triplicates were removed for quantification on indicated days. Error bars represent standard deviation. * *p* < 0.05 (Student’s T test). (**E**,**F**) SEM imaging of biofilm prevention. (**E**) Washed, uncoated PICC line surface after seven days of challenge with *P. aeruginosa*. (**F**) Coated PICC line surface after seven days of challenge with *P. aeruginosa*.

**Figure 4 ijms-24-14923-f004:**
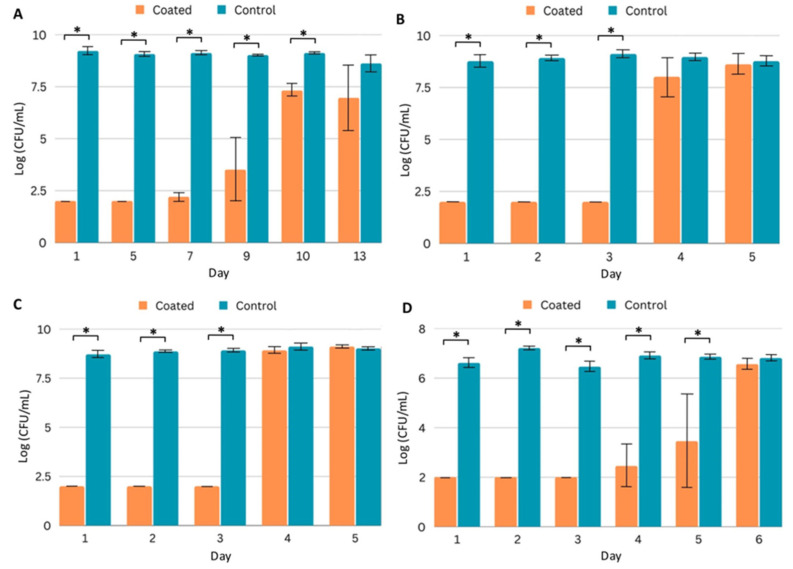
Efficacy of coated (Coating C) PICC line segments in a high-protein (70 mg/mL serum protein) environment. (**A**–**D**) Quantification of planktonic pathogen after daily challenges to control and coated PICC line segments in FBS-supplemented growth media. Pathogens quantified are (**A**) MRSA, (**B**) *P. aeruginosa*, (**C**) *K. pneumoniae*, and (**D**) *C. albicans*. Error bars represent standard deviation. * *p* < 0.05 (Student’s T test).

## Data Availability

The data presented in this study are available in this article. Further original data can be obtained from the corresponding author upon reasonable request.

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
