# Peer review of "Incorporating Ceragenins into Coatings Protects Peripherally Inserted Central Catheter Lines against Pathogen Colonization for Multiple Weeks"

_ijms, 2023, doi:10.3390/ijms241914923_

Round 1

Reviewer 1 Report

The article discusses the development of polyurethane coatings for PICC (Peripherally Inserted Central Catheter) lines to prevent colonization by common pathogens. The coatings incorporate CSA-131NDSA, and the study investigates various formulations and their antimicrobial efficacy against a range of pathogens, including MRSA, P. aeruginosa, K. pneumoniae, and C. albicans. The article provides valuable insights into the development of a coating system incorporating CSA-131 to address central line-associated bloodstream infections (CLABSIs) and other healthcare-associated infections (HAIs). The research demonstrates the potential of this coating system to prevent microbial colonization on medical devices. However, there are a few aspects that could be addressed to enhance the clarity and comprehensiveness of the article. Here are some comments and questions:

Introduction:

The introduction mentions statistics regarding HAIs and antibiotic resistance. It might be beneficial to include the source and year of the statistics to ensure that readers have access to the most up-to-date information.

The introduction introduces ceragenins and CSA-131 as potential solutions to combat antibiotic resistance. It would be helpful to provide a brief explanation of what ceragenins are and how CSA-131 works, as this information will be crucial for understanding the subsequent sections of the article.

The introduction effectively highlights the potential of CSA-131 in preventing colonization of medical devices. However, it would be beneficial to explicitly state the research gap or problem statement that the study aims to address. What specific challenges or questions is this research attempting to answer?

It would be helpful to conclude the introduction with a clear statement of the study's objectives or hypotheses. What is the research aiming to achieve in terms of coating PICC lines with CSA-131? Furthermore, I recommend that the authors compare their research objectives more explicitly with analogous studies in the field. Drawing parallels with similar research endeavors will help highlight the uniqueness and significance of their work.

In the introduction section of the article, it would greatly enhance the context and depth of the research if you could expand on the compatibility of CSA-131 with polymer coatings. Specifically, it would be valuable to provide a more comprehensive comparison with existing literature on similar antimicrobial coatings and their compatibility with different polymers. Types of Polymers: Discuss the range of polymer coatings commonly used in medical devices and the challenges associated with incorporating antimicrobial agents into these coatings. Are there specific polymers that have shown greater compatibility with antimicrobial agents in previous research?

Methodology:

Could you elaborate on the rationale behind choosing urethane and N-vinylpyrrolidone prepolymers as the starting materials for the coating?

What is the rationale for using methyl ethyl ketone as a solvent for the urethane prepolymer solution?

How was the silicone primer selected, and why was it deemed necessary for the coating process?

What criteria were used to determine the concentrations of urethane prepolymers and CSA-131NDSA in Coating B?

Figure 3. SEM imaging of biofilm prevention. (e) Washed, uncoated PICC line surface after seven days of challenge with P. aeruginosa. (f) Coated PICC line surface after seven days of challenge with P. aeruginosa. Please compare images with the same magnification.

I kindly request that you optimize Figure 5 to ensure it fits within the page constraints of the publication.

The biological research presented in this article is of significant importance and provides valuable insights into the antimicrobial efficacy of CSA-131-based coatings. To enhance the depth and comprehensiveness of this section, I strongly recommend expanding it, particularly by increasing the description of the results, and their implications. It would be beneficial to include more references to the existing literature that support and contextualize your biological research findings. This can help establish the significance of your results within the broader scientific community. To enhance the visual representation of your research, consider including more scanning electron microscopy (SEM) images for each type of bacteria tested. Visualizing the effects of CSA-131-based coatings on different bacterial species can provide valuable insights into their antimicrobial properties.

How does the performance of Coating C in preventing biofilm formation compare to other existing anti-biofilm strategies?

In order to provide a comprehensive and well-rounded research article, I would like to request the inclusion of a dedicated "Conclusion" section.

Reviewer 2 Report

Title: “Incorporating Ceragenins into Coatings Protects Peripherally Inserted Central Catheter Lines Against Colonization for Multiple Weeks “

In this work the authors aimed to generate a system of CSA-131-containing coatings for medical devices that can be adjusted to match elution and compound load for various environments and establish their efficacy in preventing growth of common pathogens in and around these devices.
More specifically, peripherally inserted central catheter lines were selected for the substrate in this work, and a polyurethane-based system was used to establish coatings for evaluation. Microbial challenges by methicillin-resistant Staphylococcus aureus, Pseudomonas aeruginosa, Klebsiella pneumoniae, and Candida albicans were performed and SEM was used to evaluate coating structure and colonization. Results indicate selected coatings have activity against selected planktonic pathogens that extend between 16 and 33 days, with similar periods of biofilm prevention

General comments: The current version of the main text should be improved to enhance the quality and the impact of the work

Some specific comments:

Figure 1. Characterization of three coating systems: Coating A samples have a 6 µm polyurethane 94
coating containing 20% (w/w) CSA-131NDSA; Coating B samples have an 18 µm polyurethane coat containing 50% (w/w) CSA-131NDSA; Coating C samples have an 18 µm polyurethane base coat containing 50% (w/w) CSA-131NDSA with a 12 µm polyurethane top coat. (a) Antimicrobial efficacy assay of Coating A against repeated inoculations with MRSA. Experiments run in triplicate. *p<05. (b) Total extraction of CSA-131 from triplicates of Coating A and Coating B. (c) Representative elution profiles of CSA-131 from Coating B and Coating C. (d) SEM image taken of uncoated PICC line surface. (e) SEM image taken of a PICC line surface coated with Coating B. Defects from handling, located along the lower edge, were included to facilitate focusing. (f) SEM image of PICC line cross section of Coating C. Frontiers between coating layers are visible and highlighted while layers are labeled for clarity. Error bars represent standard deviation.

*) This figure should be improved. The quality of the subfigures should be enhanced (increase the dpi). What is *p<05 ? Perhaps p<0.05 ? Please explain better the statistics and correct errors.

Figure 3. Efficacy of coated PICC line segments against biofilm formation. (A-D) Quantification of pathogen recovered from biofilm after daily challenges to control and coated (Coating C) PICC line segments by (a) MRSA, (b) P. aeruginosa, (c) K. pneumoniae, and (d) C. albicans. Samples were challenged with fresh inocula and media daily, and triplicates were removed for quantification on indicated days. Error bars represent standard deviation. *p<05. (e-f) SEM imaging of biofilm prevention. (e) Washed, uncoated PICC line surface after seven days of challenge with P. aeruginosa. (f) Coated PICC line surface after seven days of challenge with P. aeruginosa

*) This figure should be improved. It should be centred or re-sized. Again, what is *p<05 ? Please correct and explain better the statistical methods used.

Figure 4. Efficacy of coated (Coating C) PICC line segments in a high-protein (70 mg/mL serum protein) environment. (a-d) Quantification of planktonic pathogen after daily challenges to control and coated PICC line segments in FBS-supplemented growth media. Pathogens quantified are (a) MRSA, (b) P. aeruginosa, (c) K. pneumoniae, and (d) C. albicans. Error bars represent standard deviation. *p<.05.

*) This figure could be improved and, again what is *p<.05 ? Please provide statistical methods.

*) A section “Conclusion” is currently lacking.. please insert within the novel version of the main text.

The language could be improved

Reviewer 3 Report

Please document attached

Minor editing of English language required

Round 2

Reviewer 1 Report

The article can be accepted in the present form.

Reviewer 3 Report

The authors have addressed all of my queries and I am happy for it to be accepted with minor English revision.

 Minor editing of English language required